# Microbiome Responses to Fecal Microbiota Transplantation in Cats with Chronic Digestive Issues

**DOI:** 10.3390/vetsci10090561

**Published:** 2023-09-06

**Authors:** Connie A. Rojas, Zhandra Entrolezo, Jessica K. Jarett, Guillaume Jospin, Dawn D. Kingsbury, Alex Martin, Jonathan A. Eisen, Holly H. Ganz

**Affiliations:** 1Genome Center, University of California, Davis, CA 95616, USA; connie@animalbiome.com (C.A.R.); jaeisen@ucdavis.edu (J.A.E.); 2Department of Evolution and Ecology, University of California, Davis, CA 95616, USA; 3AnimalBiome, Oakland, CA 94609, USA; zhandra@animalbiome.com (Z.E.); jess@animalbiome.com (J.K.J.); guillaume@animalbiome.com (G.J.); alex@animalbiome.com (A.M.)

**Keywords:** FMT, fecal transplant, fecal microbiome, gut microbiome, dysbiosis, diarrhea, vomiting, antibiotics, domestic cats

## Abstract

**Simple Summary:**

The effectiveness and potential impacts of fecal microbiota transplants (FMT) on the intestinal microbial communities of domestic cats have been severely understudied. To date, only two case studies have examined FMT treatment in resolving diarrhea and chronic colitis in two adult cats. Here we provide an analysis of the fecal microbiome responses to an oral capsule FMT course in a group of 46 cats experiencing chronic vomiting, diarrhea, and/or constipation. Changes in the composition of the fecal microbiome were observed for all cats. Fecal microbiome responses were correlated with clinical signs and dry kibble consumption. Furthermore, we compared the fecal microbiomes of FMT recipients to those from their stool donors (N = 10) and from cats in a healthy reference set (N = 113) and discussed findings regarding donor bacterial engraftment in FMT recipients. We also report increases in the fecal microbiome similarity between FMT recipients and healthy cats. Our study attempts to address a large gap in the literature and provides a comprehensive analysis of fecal microbiome changes in a cohort of cats receiving oral FMTs.

**Abstract:**

There is growing interest in the application of fecal microbiota transplants (FMTs) in small animal medicine, but there are few published studies that have tested their effects in the domestic cat (*Felis catus*). Here we use 16S rRNA gene sequencing to examine fecal microbiome changes in 46 domestic cats with chronic digestive issues that received FMTs using lyophilized stool that was delivered in oral capsules. Fecal samples were collected from FMT recipients before and two weeks after the end of the full course of 50 capsules, as well as from their stool donors (N = 10), and other healthy cats (N = 113). The fecal microbiomes of FMT recipients varied with host clinical signs and dry kibble consumption, and shifts in the relative abundances of *Clostridium*, *Collinsella*, *Megamonas*, *Desulfovibrio* and *Escherichia* were observed after FMT. Overall, donors shared 13% of their bacterial amplicon sequence variants (ASVs) with FMT recipients and the most commonly shared ASVs were classified as *Prevotella* 9, *Peptoclostridium*, *Bacteroides*, and *Collinsella*. Lastly, the fecal microbiomes of cats with diarrhea became more similar to the microbiomes of age-matched and diet-matched healthy cats compared to cats with constipation. Overall, our results suggest that microbiome responses to FMT may be modulated by the FMT recipient’s initial presenting clinical signs, diet, and their donor’s microbiome.

## 1. Introduction

Although not an exact proxy, the fecal microbiome reflects the trillions of microbes that reside in the gastrointestinal tract; microbes that collectively contribute to host digestion, immunity and pathogen defense, and intestinal barrier homeostasis [1]. While the composition of the fecal microbiome is highly dynamic and can correlate with a range of host factors including age [2,3], sex [4], body condition [5], diet [6,7], and antibiotic use [8], more persistent changes in composition have also been associated with disease and infections [9,10]. Compared to their healthy counterparts, the fecal microbiome of animals with a disease or infection may have reduced microbial diversity [11], decreased abundances of functionally important microbes like fermentative or short-chain fatty acid (SCFA)-producing bacteria [12], or elevated abundances of pathogenic taxa [13,14]. These perturbations to fecal microbiome composition can directly impact host health. Recent research shows that fecal microbiota transplants (FMTs), which involve the transfer of fecal microbes from a healthy donor into the gastrointestinal tract of a recipient animal, can potentially treat a range of health conditions and improve health outcomes [15,16,17]. FMTs may be able to repopulate the microbiome and restore fecal microbiome composition by increasing microbiome diversity, enhancing the numbers of beneficial microbes and their metabolites, promoting synergistic microbiome-microbe interactions, or outcompeting pathogens [15,16,17]. FMTs have been used in lieu of conventional treatment methods but can also supplement already existing treatment methods to resolve clinical signs. Furthermore, compared to prebiotics or probiotics, FMT allows the transfer of a complex community of microbes including beneficial commensal bacteria, fungi, and bacteriophages [18]; the vast majority of which are not found in probiotics today. This might make FMT a more suitable method than prebiotics or probiotics for certain conditions [19].

FMT procedures have been used in veterinary practice since at least the 18th century to treat cattle, horses, sheep, and other animals suffering from rumination disorders, indigestion, inappetence, and colitis [20,21]. More recent efforts have used FMTs to treat acute diarrhea, relapsing chronic diarrhea, canine parvovirus, and chronic enteropathies in dogs [22,23,24,25]. Fecal transplants can also be administered prior to pathogen exposure or disease onset to prevent a condition [16]. In nursing pigs, individuals given fecal material from healthy, high-parity sows experienced reduced mortality, increased weight gain, and sustained production of antibodies when infected with porcine circoviruses compared to mock-transplanted controls [26]. Similarly, dogs given maternal fecal inoculum during the weaning period demonstrated a decreased incidence of diarrhea compared to dogs who did not receive FMTs [27]. These lines of evidence suggest that fecal transplants have the potential to lead to improved health outcomes in animals.

While there is growing interest in the application of FMTs in people, particularly to treat recurring antibiotic-resistant *Clostridioides difficile* infections [28,29], there are few published studies that have examined the effects of FMT on the fecal microbiomes of companion animals. Prior research is mostly limited to case studies of FMT treatment in a single individual or a few individuals, but larger-scale analyses are missing, particularly for felids. In case studies of domestic cats (*Felis catus*), FMT treatment led to long-term resolution of vomiting and diarrhea in a six-year-old cat [30] and successfully resolved chronic ulcerative colitis in an adult cat [31].

Here, we expand on this work and use 16S rRNA gene sequencing to examine fecal microbiome responses to an oral capsule FMT course in 46 domestic cats suffering from chronic digestive conditions, including vomiting, diarrhea, and/or constipation (Table 1). We describe detailed changes observed in the fecal microbiomes of FMT recipients and determine whether these changes were correlated with four host factors (reported clinical signs, response to FMT, recent antibiotic use, and kibble consumption) (Figure 1A). Then we identify the bacterial amplicon sequence variants (ASVs)—the most refined level of taxonomy behind species—that engrafted or were shared between FMT recipients and their stool donors (Figure 1B). Lastly, we evaluate the extent to which fecal microbiomes of FMT recipients resemble those of healthy cats after the conclusion of FMT (Figure 1C). Our study addresses a large gap in the literature and evaluates fecal microbiome shifts as a response to FMT in a cohort of cats, and compares the fecal microbiomes of FMT recipients to their stool donors and to a set of healthy pet cats.

## 2. Materials and Methods

*Study animals, sample collection, and surveys.* We used social media (primarily Facebook, Instagram, and Twitter) to recruit people with cats exhibiting symptoms of a chronic digestive condition (e.g., diarrhea, vomiting, and/or constipation >2 weeks) who were interested in adding oral FMT capsules to their care. All participants (N = 46) signed an informed consent form and were sent a pilot study kit. Kits contained 50 FMT capsules, a health survey, and materials to collect two fecal samples. Participants gave one to two capsules to their cat orally with food daily for ~25 days. Some cats could tolerate the two capsules daily, and others could tolerate one capsule daily, but all cats must have completed the 50-capsule course to be part of this study. Owners were asked to collect fecal samples from their cats before and two weeks after the end of FMT to accurately assess changes in fecal microbiome composition.

To collect demographic, physical, and lifestyle information on each cat, owners filled out health surveys that asked about their cat’s age, body condition, breed, sex, spay or neuter status, diet (including veterinary diets), and any diagnoses the cats had received from veterinarians. Owners also recorded the consistency of their cat’s feces prior to beginning the capsules and following FMT (Appendix A), using a fecal scoring scale ranging from 1 (hard and dry) to 7 (watery diarrhea), with 3 and 4 considered a normal consistency [32]. In addition, owners provided photos of the fecal samples at both time points allowing us to confirm fecal scoring. Importantly, owners recorded the specific clinical signs they were hoping to alleviate with the FMT capsules and among the most common clinical signs were diarrhea, constipation, vomiting, and lack of appetite. Cats did not take any oral antibiotics during the study, but some had taken antibiotics during the twelve months preceding the study. For the majority of cats, their diet remained constant during the study period. Following the course of FMT capsules, owners were asked to describe their cat’s response to the FMT capsules, specifically, whether there was an improvement, no observable change, or a worsening of their conditions. This evaluation was based principally on fecal consistency scores and incidences of vomiting. See Table 1 for a summary of the characteristics of FMT recipients.

The study protocol and informed consent forms followed the Animal Welfare Act. The capsules were given in addition to the standard of care offered by their veterinarian for their condition. Participants were advised to consult with their veterinarian before participating in the study and they were informed that they could drop out of the study for any reason at any time.

*Preparation of FMT capsules.* Donated fecal matter was collected from ten healthy indoor cats: from six individual donors and two pairs of donors (where the fecal material of two cats who lived together was combined). Donor cats had no antibiotic treatment in the past year, were not taking medications, had no known health conditions, current infections, or recent surgeries, and did not exhibit behavioral issues. The compositions of their fecal microbiomes were comparable to those of healthy cats in our reference database, as determined by 16S rRNA gene sequencing. All donor samples that had appropriate fecal consistencies (scores of 3–4) were submitted for pathogen screening using both qPCR and culturing to the University of California, Davis Real-time PCR and Diagnostics Core Facility. Samples were screened for *Clostridiodes difficile* toxins A and B, *Cryptosporidium* spp., *Salmonella* spp., *Giardia* spp., feline coronavirus, feline parvovirus (Panleukopenia), and *Tritrichomonas foetus*. Donors were also periodically screened for helminth parasites and protozoan oocysts via fecal flotation (IDEXX). Donors averaged 4.93 ± 3.43 years of age and were mostly domestic shorthairs (75%) that consumed a diet of both wet food and kibble (50%; Appendix A).

Fecal samples from these donors served as material for the FMT capsules (Appendix A) that were administered to recipients. Any stool with a suboptimal fecal consistency, excess mucus, mold, flies, unpleasant odor, or color was discarded. The stool was cleaned of litter, hair, and any foreign material prior to mixing with 20% glycerol (wet weight) and freeze drying at −40 °C to −50 °C. Once the stool was completely freeze-dried, it was stored at −20 °C until it was ground into a fine powder and placed into size 4 capsules. We followed stringent quality control guidelines and standard operating procedures. Capsule batches were labeled by donor and lot number, and randomly assigned to FMT recipients.

Using data from Ganz et al., 2022 [3] (Appendix A); we compared a reference set of 113 healthy cats for comparison with FMT recipients. These cats had no known health conditions; clinical signs or diagnoses; had a body condition score between 4 and 6 (mean 5.15 ± 0.5); and had taken no antibiotics within the previous 12 months according to information reported by their owners. The cat’s ages were between 1–12 years of age (mean 5.4 ± 3.26 years of age); although for statistical analyses, FMT recipients were age-matched and diet-matched to cats from the healthy reference set (see ‘*Statistical analyses: fecal microbiome similarity between FMT recipients and healthy pet cats*’). Of these 113 cats, 56% were female, about half were domestic shorthairs (49%), and almost all were spayed or neutered (92%) (Appendix A). Seventy percent of healthy cats incorporated dry kibble into their diet, 66% consumed some amount of canned food, and 28% included raw food in their diet.

*DNA Extractions and 16S rRNA gene sequencing.* Owners placed fecal material from their cats (n = 46) pre- and post-FMT into 2 mL screw cap tubes containing 100% ethanol and silica beads. These were shipped to AnimalBiome’s facilities (Oakland, CA) and stored at 4–8 °C until further laboratory analysis. Fecal material was isolated from the preservation buffer by centrifugation, and genomic DNA was extracted using QIAGEN DNeasy PowerSoil Kits (Germantown, MD). Briefly, samples were placed in bead tubes containing C1 solution, incubated at 65 °C for 10 min, and placed in a bead beater for 2 min, after which the manufacturer’s protocol was followed as written. DNA was extracted from the fecal samples of healthy cats (n = 113) and donors (n = 20) in the same fashion. For the donors, four cats had one fecal sample each, one cat had two samples, and another cat had 6 samples. The two pairs of donors (where two cats cohabited and their fecal material was combined), had three and five fecal samples, respectively. Thus, 20 fecal samples total were collected from 10 donors for microbiome analysis.

Amplicon libraries of the V4 region of the 16S rRNA gene were generated from extracted DNA using a dual-indexing one-step PCR with complete fusion primers (505F/816R) (Ultramers, Integrated DNA Technologies) with multiple barcodes (indices), adapted for the Illumina Miniseq platform as detailed in Pichler et al., 2018 [33]. PCR reactions contained 0.3–30 ng template DNA, 0.1 µL Phusion High-Fidelity DNA Polymerase (ThermoFisher), 1X HF PCR Buffer, 0.2 mM each dNTP, and 10 µM of the forward and reverse fusion primers. PCR conditions were as follows: denaturation at 98 °C for 30 s, 30 cycles of 98 °C for 10 s, 55 °C for 30 s, and 72 °C for 30 s. There was a final incubation at 72 °C for 4 min 30 s for a final extension and hold at 6 °C until retrieval. PCR product amplification was assessed by running on 2% E-Gels with SYBR Safe (ThermoFisher). PCR products were then purified and normalized using the SequalPrep Normalization Kit (Thermo Fisher). PCR products were pooled into the final libraries; each contained 95 samples (not all from this study) and 1 negative control (blank sample extraction). The final libraries were quantified with QUBIT dsDNA HS assay (Thermo Fisher), diluted to 1.8 pM and denatured according to Illumina’s specifications for the MiniSeq. Identically treated phiX was included in the sequencing reaction at 15%. Paired-end sequencing (150 bp) was performed on one mid-output Illumina MiniSeq flow cell per 96-well plate of samples with positive and negative controls.

*Processing 16S rRNA gene sequences.* Raw 16S rRNA gene amplicon sequences were obtained from fecal samples of FMT recipients (46 individuals, 92 fecal samples), donors (10 individuals, 20 fecal samples), and cats in the healthy reference set (113 individuals, 113 fecal samples [3]). Sequences were trimmed, quality-filtered and dereplicated using the Divisive Amplicon Denoising Algorithm (DADA2 v1.14.1) pipeline in R (v3.6.2) [34,35]. Specifically, forward reads were trimmed to 145 bp while reverse reads were trimmed to 140 bp. After calculating error rates, ASVs were inferred using DADA2’s core denoising algorithm and chimeras were removed. The DADA2 pipeline merges paired-end reads after ASV calling to achieve greater accuracy [34]. Overall, samples from cats receiving FMT contained an average of 52,615 ± 26,134 sequences after processing in DADA2, donors averaged 62,299 ± 23,558 sequences, and those from the healthy reference set contained on average 45,786 ± 27,918 sequences. Taxonomic classifications were assigned to ASVs using DADA2′s naïve Bayesian classifier against the Silva reference database (v138) [36,37] and ASVs classified as Eukarya, Chloroplasts, or Mitochondria were removed from the dataset. Overall, the recipient fecal microbiome dataset contained 2877 total ASVs, the donor dataset contained 2060 ASVs, and the healthy reference dataset had 2566 ASVs. The ASV relative abundance table, list of ASV taxonomic assignments, and metadata for fecal samples from recipients, donors, and healthy cats are available as Appendix A.

*Statistical analyses: defining host explanatory variables.* Unless otherwise stated, all statistical analyses and figures were performed in R (v3.6.2) [35]. Throughout analyses, samples were classified as “recipient” samples if they came from cats that participated in the FMT capsule course, “healthy” samples if they came from the 113 healthy cats, and “donors” if they came from any of the 10 stool donors. “Recipient” samples were further categorized into “preFMT” or “postFMT” depending on when they were collected.

For recipient samples, we conducted various statistical analyses that involved four host predictors of interest: response to FMT capsules, initial reported clinical signs, prior antibiotic use, and dry kibble consumption (Figure 1A). For the ‘response to FMT’ variable, recipients were categorized as “Responders” if their owners reported improvement in their clinical signs following FMT, or as “Non-Responders” if their owners reported no change or a worsening of clinical signs following FMT. This categorization was further corroborated by analysis of changes in fecal scores. Clinical signs were categorized into Diarrhea only, Constipation only, Vomiting with Diarrhea, and Vomiting with Constipation (Table 1). Cats that were previously on antibiotics were categorized as “Yes”, and all other cats as “No”. Lastly, for the diet category, we compared cats that included dry kibble in their diet vs. cats that did not incorporate dry kibble in their diet. Table 1 lists sample sizes for each host factor and its subcategories. We also had information on a cat’s breed, body condition score, spay or neuter status, etc., but these were not selected as variables for analysis because there was not sufficient variation among samples (Table 1). For example, 100% of cats had been sterilized, so this variable was not appropriate for statistical analysis.

*Statistical analyses: fecal microbiome alpha-diversity and beta-diversity*. We calculated the fecal microbiome alpha-diversity of samples from FMT recipients. We did not rarefy samples by subsampling to an equal number of sequences as this has been shown to be statistically inappropriate for microbiome analyses [38]. All samples had at least 8600 reads. ASV richness was calculated with phyloseq (v1.30.0) [39], Pielou’s evenness with microbiome (v1.15.2) [40], and Faith’s Phylogenetic Diversity with picante (v1.8.2) [41]; the latter after supplying it with a phylogenetic tree of ASV sequences built with DECIPHER (v2.14.0) [42] and phangorn (v2.5.5) [43].

Linear mixed models evaluated whether FMT recipients varied in their fecal microbiome richness, evenness, or phylogenetic diversity, using the lme4 (v1.1-29) [44] package. The models included one of the three alpha-diversity metrics as the dependent variable, regressed preFMT (or postFMT) alpha-diversity values against the four host predictors (response to FMT, prior antibiotic use, clinical signs, and kibble consumption), and set host age and sex as random effects (Figure 1A). The statistical significance of the independent variables was determined via likelihood ratio tests (LRT) using the car package (v3.0-13) [45], setting a = 0.05. Boxplots of fecal microbiome alpha-diversity were made in ggplot2 (v3.3.6) [46].

To analyze fecal microbiome beta-diversity in FMT recipients, we constructed two types of distance matrices from unrarefied data using phyloseq. The two distances were Bray–Curtis dissimilarity index, which captures compositional dissimilarity between samples, and Aitchison distance, which is thought to be more appropriate for compositional data. For this, singleton and doubleton ASVs were removed from the dataset and ASV counts were converted to proportions for Bray–Curtis distances or applied a center-log ratio transformation for Aitchison distances. We constructed marginal Permutational Multivariate Analysis of Variance (PERMANOVA) models using the vegan package (v2.6-2) [47] to examine whether FMT recipients varied in their fecal microbiome structure (e.g., beta-diversity). The models included one of the two distances as the dependent variable, used 999 permutations, and regressed preFMT (or postFMT) beta-diversity against the four host predictors, plus two more variables we needed to account for (age, and sex) (Figure 1A). If PERMANOVA tests yielded statistically significant relationships (a = 0.05), we used pairwise PERMANOVA tests to discern which groups were different, and Permutational Analyses Of Multivariate Dispersions (PERMDISP) [47] to test for differences in microbiome dispersion between groups. Principal Coordinates Analysis (PCoA) ordinations were made in ggplot2.

*Statistical analyses: change in the relative abundances of bacterial genera.* We investigated whether the relative abundances of core or potentially pathogenic taxa changed in FMT recipients and whether these changes were associated with four host predictors. We defined core genera (N = 21) as the genus-level bacterial taxa that were found in at least 55% of samples from a healthy population of cats [3] (Appendix A). The potentially pathogenic taxa were six bacterial genera that may be pathogenic to felines (Appendix A) [48,49,50,51]. All taxa tested must have been found in at least 10% of samples from FMT recipients. The generalized linear models specified change (Δ) in the relative abundance of bacterial taxon *i* (postFMT–preFMT) as the dependent variable and four host predictors as independent variables: response to FMT, prior antibiotic use, clinical signs, and kibble consumption. These models did not include host sex or age as predictors since neither significantly predicted fecal microbiome alpha- or beta-diversity. We used the stats package [35] for the linear models and statistical significance was evaluated using likelihood ratio tests as described earlier. If the term in the model was statistically significant (*p* < 0.05), we followed up with multiple comparison testing using the multcomp R package [52] and reported Tukey-adjusted *p*-values. Plots of the change (Δ) in relative abundance for statistically significant genera were constructed using ggplot2.

*Statistical analyses: ASVs shared between FMT recipients and their stool donors*. The second objective of our study was to identify which donor microbes could engraft (i.e., transfer) in FMT recipients and calculate ASV sharing rates between FMT recipients and their stool donors (Figure 1B). We used calculations adapted from a meta-analysis by Ianiro et al. 2022 [53], which examined strain engraftment after FMT in human cohorts across eight different diseases. In our study, ASV sharing rates were calculated by dividing the number of ASVs shared between postFMT samples and their stool donors (excluding taxa shared between preFMT samples and donors) by the total number of ASVs in the donor sample (excluding any taxa shared with preFMT samples). A linear mixed model evaluated whether an FMT recipient’s ASV engraftment rate correlated with the four host predictors of interest and set host age and sex as random effects. Another generalized linear model tested whether ASV engraftment rates varied among stool donors. Plots of ASV sharing rates were made in ggplot2.

*Statistical analyses: fecal microbiome similarity between FMT recipients and healthy pet cats.* Lastly, we tested whether the fecal microbiomes of FMT recipients approximated or “became more similar” to the fecal microbiomes of age-matched and diet-matched healthy cats (Figure 1C) [3]. We wanted to investigate for example, whether the fecal microbiomes of cats that had been exposed to antibiotics became more similar or less similar to healthy cats that had not had antibiotics. For these analyses, we used the two distances constructed for beta-diversity analysis and only kept pairwise comparisons between recipients and cats in the healthy reference set (e.g., we excluded recipient vs. recipient comparisons or healthy vs. healthy comparisons). Only pairwise comparisons of cats whose ages were within 1 year of each other were retained, and dyads must have had matching diets in terms of their dry kibble consumption (e.g., if the FMT recipient ate dry kibble, so must have the healthy cat). Change (Δ) in fecal microbiome similarity was calculated as follows:

[similarity between postFMT sample for cat *i* and healthy animal *ii*]−

[similarity between preFMT sample for cat *i* and healthy animal *ii*].

These calculations were conducted for all 46 FMT recipients compared against their age-matched kibble-matched healthy counterparts, and the resulting data frame was used for statistical analysis. A generalized linear model tested whether the change (Δ) in fecal microbiome similarity was correlated with the four host predictors. Statistical significance was evaluated using likelihood ratio tests (LRT) as described earlier. Plots of average Δ in fecal microbiome similarity (Bray–Curtis) were constructed using ggplot2.

## 3. Results

### 3.1. Characteristics of FMT Participants

A total of forty-six cats underwent oral capsule FMTs and provided data for this study. They were 56% male, ranged in age from 1 to 10 years of age (average 10.23 yrs old), and were mostly domestic shorthairs (74%). They were all spayed or neutered and fifty-two percent had been on antibiotics within the twelve months preceding the beginning of FMT (although none took antibiotics during the study period). Prior to FMT, 41% of FMT recipients had been experiencing diarrhea only, 17% of recipients experienced only constipation, 33% experienced vomiting with diarrhea and 9% experienced vomiting with constipation.

### 3.2. Examining Microbiome Variation before and after FMT

A principal goal of our study was to document whether shifts in the microbiome were observed after FMT. Examination of the fecal microbiome compositions of participants (Figure 2 and Appendix A) revealed that the relative abundances of bacterial genera did shift with FMT in some participants. Some cats experienced decreases in their *Blautia* or *Collinsella* relative abundances while others showcased increases in their *Prevotella* 9, *Megasphera*, and *Megamonas* relative abundances (Figure 2 and Appendix A). For some cats, however, the relative abundances of predominant bacterial genera were constant between the two timepoints (Figure 2 and Appendix A).

Next, we correlated fecal microbiome alpha- and beta-diversity pre- and postFMT with four main host factors of interest: response to FMT, initial clinical signs, prior antibiotic use, and dry kibble consumption (Figure 1A). Results showed that before FMT, the fecal microbiome alpha-diversity of FMT recipients was best predicted by clinical signs and dry kibble consumption (*p* < 0.05, see Appendix A). Cats that experienced vomiting and diarrhea had less diverse fecal microbiomes than cats that suffered from vomiting and constipation (Figure 3A). Cats that incorporated any amount of dry kibble in their diet tended to harbor less diverse fecal microbiomes than cats that did not consume dry food (*p* < 0.05, Figure 3B). After FMT, fecal microbiome diversity was also significantly correlated with an individual’s pre-FMT clinical signs (*p* < 0.05, see Appendix A). The bacterial communities of cats experiencing vomiting with diarrhea were less even than cats who only had diarrhea.

Fecal microbiome beta-diversity before and after FMT was similarly best predicted by clinical signs and dry kibble consumption, which accounted for 9% and 3% of the variation in fecal microbiomes among individuals, respectively (*p* < 0.05, Figure 3C,D, see Appendix A). Generally, differences laid between the fecal microbiomes of cats experiencing constipation (or vomiting with constipation) and the fecal microbiomes of cats with diarrhea (or vomiting with diarrhea) (pairwise PERMANOVA, *p* < 0.05; all other pairwise comparisons *p* > 0.05). The patterns were similar for both pre-FMT samples and post-FMT samples. On a PCoA ordination, a clear distinction is observed between the fecal microbiomes of cats with diarrhea (and diarrhea with vomiting) and those of cats with constipation (Figure 3C). Clear separation of clusters was also observed for the fecal microbiomes of cats that consumed dry kibble vs. those that did not (Figure 3D). Response to FMT, prior antibiotic use, age (years), or sex were not significantly associated with fecal microbiome variation (*p* > 0.05, see Appendix A).

The fecal microbiome dispersions of hosts before FMT did not vary with host clinical signs (PERMDISP Bray–Curtis *F* = 1.63, *p* = 0.19; Aitchison *F* = 1.69, *p* = 0.18) but did vary with host dry kibble consumption when utilizing Aitchison distance as our metric (PERMDISP Bray–Curtis *F* = 0.10, *p* = 0.74; Aitchison *F* = 6.43, *p* = 0.14). Thus, microbiome compositional differences between groups could be partially attributed to differences in their microbiome dispersions. Microbiomes post-FMT varied in their dispersions when comparing hosts of different clinical signs (PERMDISP Bray–Curtis *F* = 0.96, *p* = 0.41; Aitchison *F* = 3.64, *p* = 0.02), but not between hosts with different dry kibble preferences (PERMDISP Bray–Curtis *F* = 0.44, *p* = 0.5; Aitchison *F* = 1.16, *p* = 0.28).

### 3.3. Changes in the Relative Abundances of Core and Pathogenic Bacterial Genera in FMT Recipients

Next, we conducted more fine-scale analyses to identify bacterial taxa that may have decreased or increased in abundance after FMT. Specifically, we tested whether the change in relative abundance (postFMT–preFMT) of core genera or potentially pathogenic genera was correlated with host response to FMT (Responder vs. Non-Responder), initial clinical signs, prior antibiotic use, and dry kibble consumption.

Of the 21 bacteria genera that constituted the core in the fecal microbiomes of healthy pet cats [3] (Appendix A), the relative abundances of four genera (*Clostridium*, *Collinsella*, *Negativibacillus* and *Subdoligranulum*) were associated with clinical signs. Generally, the relative abundances of *Collinsella* and *Negativibacillus* tended to increase in cats experiencing vomiting with constipation relative to other cats, but the opposite was true of *Subdoligranulum* relative abundances (Figure 4A–D, Appendix A, see Appendix A for posthoc comparisons). The relative abundances of another four bacterial genera (*Butyricoccus*, *Megamonas*, *Peptococcus*, and *Ruminococcus*) changed differentially in Responders vs. Non-Responders (Appendix A). With the exception of *Megamonas*, the relative abundances of the aforementioned bacterial groups tended to decrease slightly in Responders compared to Non-Responders (Figure 4F–I, Appendix A for posthoc comparisons). The relative abundances of *Negativibacillus* tended to increase in cats that had not previously taken antibiotics compared to cats that had a previous antibiotic exposure (*p* < 0.05, Figure 4K and Appendix A). Cats who did not incorporate any amount of kibble in their diet tended to showcase greater increases in their *Peptoclostridium* and *Subdoligranulum* relative abundances compared to cats that consumed dry kibble (Figure 4N,O and Appendix A).

When examining the change in the relative abundances of the six potentially pathogenic genera (Appendix A), FMT recipients who responded well to FMT (Responders) exhibited decreases in their *Veillonella* loads compared to Non-Responders, which mainly exhibited no change in their *Veillonella* abundances (Figure 4G and Appendix A). Cats with vomiting and constipation as their clinical sign showcased increases in their *Desulfovibrio* relative abundances compared to recipients reported to have only diarrhea or only constipation (Figure 4G and Appendix A). Lastly, the fecal microbiomes of recipients without a recent antibiotic exposure showed increases in their *Desulfovibrio* and *Escherichia* relative abundances compared to cats that had recently taken antibiotics, which instead showed decreases in the abundances of these two bacterial groups (Figure 4G and Appendix A) Diet-associated shifts in the relative abundances of potentially pathogenic genera were not observed among FMT participants (Appendix A).

### 3.4. ASVs Shared between FMT Recipients and Their Stool Donors

Another objective of our study was to compare the fecal microbiomes of FMT recipients with those of their donors to identify the bacterial ASVs that were more likely to be shared with or engraft in FMT recipients (Figure 1B). We also tested whether individuals varied in their ASV engraftment efficiency and correlated these values with host predictors. For this, we analyzed the ASVs that were shared between the fecal microbiomes of FMT recipients post-FMT and their specific stool donors, excluding any ASVs that were shared between the two groups pre-FMT. ASV engraftment rates indicated the proportion of donor ASVs that engrafted into FMT recipients relative to the total number of donor ASVs that had the capacity to engraft.

Across FMT recipients, ASV engraftment rates ranged from 3.25% to 26.14%, with an average of 12.64% (±6.16%) (Figure 5A, Appendix A). That is, of the bacterial ASVs present in FMT stool donors with the capacity to engraft or be shared ($\bar{x}$: 559 ASVs, range: 169–1086 ASVs), about 13% on average ($\bar{x}$: 64 ASVs, range: 14–172 ASVs) successfully engrafted in the FMT recipient (Appendix A). Thus, complete microbiome engraftment was not observed in this cohort of cats. An FMT recipient’s ASV engraftment rate was not significantly associated with response to FMT, initial clinical signs, prior antibiotic use, or dry kibble consumption (LMM LRT Response χ^2^ = 0.03, *p* = 0.95; Clinical signs χ^2^ = 0.55, *p* = 0.9; Antibiotics χ^2^ = 2.12, *p* = 0.14; Dry food χ^2^ = 0.05, *p* = 0.82). ASV sharing rates, however, were significantly predicted by donor’s identity (Kruskal Test χ^2^ = 15.32, *p* = 0.03), with two donors in particular (D2 and D7) having bacterial ASVs that were shared at a larger frequency than the ASVs of other donors (Figure 5B, Appendix A). The ASV engraftment rate was the lowest for three donors- D1, D4, and D5; the former two which represent donors where the fecal material of two cohabiting cats was combined to make a single donor. These donors had diverse fecal microbiomes to start with, but not higher than other donors, so it is not completely clear why much fewer ASVs engrafted from these donors.

The most commonly shared ASVs (ASVs that engrafted > 10 of the 46 FMT recipients) were classified as *Bacteroides*, *Clostridium*, *Lachnoclostridium*, *Enterococcus*, *Peptoclostridium*, *Blautia*, *Fusobacterium*, *Clostridium*, *Blautia*, unclassified *Butyricicoccaceae*, unclassified *Oscillospiraceae*, unclassified *Oscillospirales*, and unclassified *Lachnospiraceae* (Appendix A). Of all of the bacterial ASVs that engrafted across the fecal microbiomes of FMT recipients, 18.43% were classified as *Prevotella* 9, 9.38% as *Peptoclostridium*, 7.5% as *Bacteroides*, 7.23% as unclassified *Lachnospiraceae*, and 5.44% as *Collinsella* (Figure 5C). Not surprisingly, the most common genera of engrafted ASVs were also the most abundant bacterial genera across the fecal microbiomes of recipients post-FMT and pre-FMT. *Prevotella* 9 was found at a mean relative abundance of 17.32% across samples from FMT recipients, *Bacteroides* at 11.5%, *Collinsella* at 6.2%, *Blautia* at 4.39%, and *Peptoclostridium* at 4.14% (Appendix A). This indicates that FMT recipients are not necessarily gaining taxonomically novel microbes but instead ASVs that are taxonomically similar to those that they start with, albeit with some genomic variation.

### 3.5. Comparing the Fecal Microbiomes of FMT Recipients and Healthy Cats

Lastly, we investigated whether the fecal microbiomes of FMT recipients two weeks after the conclusion of FMT became similar to those of age-matched and diet-matched individuals from the healthy reference set (Figure 1C). Perhaps there were groups of cats whose fecal microbiomes shifted closer to those of healthy cats compared to other groups of cats. For this, we computed the similarity between a cat’s preFMT microbiome and a fecal microbiome from the healthy reference set, and the similarity between a cat’s postFMT microbiome and that same healthy cat’s fecal microbiome. We then subtracted the two values (post—pre) to obtain (Δ) similarity for every FMT recipient—healthy animal dyad. This change in similarity (Δ) would indicate whether a cat’s fecal microbiome overall became more (if a positive value) or less (if a negative value) similar to fecal microbiomes from the healthy reference set.

We found that the fecal microbiomes of cats who responded favorably to FMT (‘Responders’) became equally similar (mean change in similarity $\bar{x}$: 0.004) to those of healthy cats compared to Non-Responders (mean change in similarity $\bar{x}$: 0.006) (Figure 6A, Appendix A). Additionally, the fecal microbiomes of cats with constipation (mean change in similarity $\bar{x}$: −0.04) became less similar to those of the healthy reference set than cats who experienced other symptoms ($\bar{x}$ 0.014 for diarrhea, $\bar{x}$ 0.007 for vomiting with diarrhea, and $\bar{x}$ 0.001 for vomiting with constipation) (Figure 6B, Appendix A). The fecal microbiomes of cats that had previously taken antibiotics become equally similar to those of healthy cats (mean change in similarity $\bar{x}$: 0.005) compared to cats who had not taken any antibiotics (mean change in similarity $\bar{x}$: 0.006) (Figure 6C). Lastly, cats that did not consume dry kibble had fecal microbiomes that became more similar to those of healthy cats ($\bar{x}$ 0.019) compared to cats that ate dry food ($\bar{x}$ 0.001) (Figure 6D, Appendix A).

## 4. Discussion

### 4.1. Host Predictors of Fecal Microbiome Alpha- and Beta-Diversity

The fecal microbiomes of FMT recipients varied depending on the initial clinical signs, and diet of the individual, with noticeable differences observed between cats fed kibble vs. cats that were not fed kibble. A dry kibble diet often contains lower amounts of protein and higher amounts of carbohydrates compared to commercial canned food or raw-food based diets. Our findings are consistent with those of a prior study conducted in dogs, which reported that individuals fed a raw meat diet exhibited more diverse fecal microbiomes than individuals fed commercial foods [54]; although other studies report a contrasting finding [55]. Diet is a major determinant of fecal microbiome composition in cats, given that microbes that are able to metabolize the dietary components outcompete those that do not. The type of food a cat is eating (raw food vs. canned wet food vs. dry kibble) will be selected for distinct microbiome compositions depending on its protein, fat, carbohydrate, and fiber [56,57,58,59]. Similar to our study, prior studies conducted in both dogs and cats report differences in the fecal microbiomes of individuals who eat kibble vs. canned wet food or kibble vs. raw foods [3,57].

We also found that cats with vomiting and diarrhea had less diverse microbiomes than cats experiencing other clinical signs. Host clinical signs in general explained 9% of the variation in fecal microbiomes among individuals, with particular differences observed between cats that suffered from constipation and cats that suffered from diarrhea. Vomiting, diarrhea, and constipation are usually not standalone symptoms and may reflect distinct disorders or diseases. Diarrhea, for example, is quite common in felines and may be a symptom of bacterial, protozoan or viral infection [60]. It may also be a clinical sign of inflammatory bowel disease (IBD) or low-grade lymphoma [61]. Vomiting is also extremely common in cats [62] and may be caused by food intolerance, inflammatory bowel disease, liver disease, pancreatitis, hyperthyroidism, and uremia [63,64,65]. Historically, constipation in cats has been tied to dehydration and is associated with chronic kidney disease (CKD), diabetes mellitus and hyperthyroidism [66]. It is evident that the clinical symptoms exhibited by cats are strongly tied to their health and physiology, which may partially explain the differences in their fecal microbiomes. Cats with diarrhea compared to cats with constipation or vomiting may have slightly different gastrointestinal physiologies, digestion, gut transit times [67], or overall health. They may have taken distinct antibiotics or medical treatments to treat their symptoms.

Interestingly, fecal microbiome composition did not vary with prior antibiotic use, that is, cats that had previously taken antibiotics did not appear to possess fundamentally distinct microbiomes than cats that had not been exposed to antibiotics. Cats that had been on antibiotics previously, however, did experience decreases in the relative abundances of *Escherichia* and *Veillonella* compared to cats not on antibiotics. This is significant given that high *Escherichia* or *Veillonella* loads can have health consequences in cats. This finding also suggests that prior antibiotic use might not affect the relative abundances of all bacterial taxa in the same way. A plethora of studies have shown associations between fecal microbiome composition and antibiotic use in companion animals [68,69,70]. Other studies have also highlighted how FMT treatment might restore fecal microbiome composition after antibiotic-associated disruptions [71], but here we are discussing how FMT may operate differently in individuals exposed to antibiotics vs. those that were not. Oral decontamination with antibiotics appears to enhance the efficacy of FMT in human patients who are colonized by beta-lactamase producing *Enterobacteriaceae* or *Acinetobacter* [72,73]. However, these types of findings are unknown for other types of bacteria and other study systems like cats.

Lastly, we found that fecal microbiome responses to FMT were highly individualized. Plots of microbiome composition clearly highlighted how microbiome responses did not look the same across FMT recipients. There is a multitude of factors that could be modulating a cat’s response to FMT, including their genetics, length and severity of their clinical signs, dietary changes, frequency and number of oral antibiotic courses, non-GI health conditions, age, sex, breed, genetics, and hormones. Thus, we encourage veterinarians, pet owners, and researchers in this field to keep these factors in mind as they consider FMT oral capsules for cats.

### 4.2. ASV Engraftment Rates in FMT Recipients

Our study also identified the donor microbes that ‘engrafted’ in the recipient following the FMT capsule course. We did so by quantifying the degree of ASV sharing between recipients and their stool donors. We found that donors shared about 13% of their ASVs (range 3–26%) with FMT recipients after FMT (excluding ASVs shared between donors and FMT recipients before FMT, which ranged from 54 to 309 ASVs). This finding indicates that complete microbiome engraftment is not occurring in this group of cats, and nor may it be necessary for FMT to be helpful. Not many FMT studies have quantified engraftment rate, but a prior study reported that in humans with HIV taking antiretroviral medication, modest microbiome engraftment was observed after FMT administration via a colonoscopy. The recipient’s fecal microbiota remained significantly distant from donors eight weeks after FMT [74]. A small clinical trial of 12 patients who had mild to moderate ulcerative colitis reported that the proportion of bacteria transferred from the donor to the FMT recipient varied from 15% to 85% [75]. The transferred bacteria spanned the phylogenetic diversity of the donor’s bacteria.

Furthermore, while we found that ASV engraftment rates were not correlated with host factors such as diet, clinical signs, or antibiotic use, these were strongly tied to donor identity. In other words, ASV engraftment rates depended on the fecal microbiome of donors. This echoes prior work which showed that strain engraftment for patients with recurrent *C. difficile* (rCDI) infection was largely predicted by the abundance and phylogeny of bacteria in the donor and the bacteria already present in the FMT recipient [76]. Another study examining FMT treatment in patients with rCDI found that donor-derived strains constituted a larger fraction of the post-FMT microbiota than did novel undetected strains not present in donors or recipients [77]. These findings reinforce the notion that potential FMT stool donors need to be rigorously screened, given the large impact they could have on the fecal microbiomes of FMT recipients. However, we cannot discount the potential influences of the FMT recipient as well. A meta-analysis examined strain engraftment in the gut of human patients across eight disease types, including rCDI, irritable bowel syndrome, Crohn’s disease, renal carcinoma, and Tourette’s syndrome. They reported that the most important predictors of strain-level retention were the diversity and the abundance of bacterial species in FMT recipients [78]. A similar meta-analysis using some of these same human cohorts found that increased engraftment was observed in individuals who received FMT from multiple routes (e.g., via both capsules and colonoscopy) [53]. Increased engraftment was also observed in individuals who had been treated with antibiotics for their infections or disease [53]. Furthermore, as Danne and colleagues (2022) [79] note, an FMT recipient’s genetics, immunity, microbiota, and lifestyle, may also impact bacterial engraftment and clinical efficacy.

We found that the most commonly shared ASVs belonged to the genera *Prevotella*, *Collinsella*, *Bacteroides*, and *Peptoclostridium*. Interestingly, all of these genera form part of the core microbiome in healthy pet cats [3]. These four bacterial genera are also correlated with SCFA (propionate, acetate, butyrate) production in the mammalian intestine [80]. SCFAs act as indispensable sources of energy for host colonocytes, stimulate colonic blood flow and motility, and may promote the growth of commensal and resident bacteria [81,82,83]. There is also increasing evidence that gut microbial metabolites like SCFAs may act as regulators of gene expression or as signaling molecules [84]. Thus, it appears that potentially beneficial microbes are being shared between stool donors and FMT recipients in our cohort of cats. In a study examining strain engraftment in humans treated with FMT for rCDI, the fecal microbiomes of FMT recipients were also enriched in *Bacteroides* (and *Alistipes* and *Parabacteroides*) after FMT [85].

## 5. Limitations

Our study has several limitations and we advise readers to interpret our findings with caution. First, the data on FMT effectiveness was solely derived from information provided by pet owners, which may be subject to biases and inaccuracies. Furthermore, our study lacked a placebo group and we encourage future studies to include a placebo group in the study design to evaluate whether oral FMTs improve clinical signs in cats. Nonetheless, even with these limitations, our study provides valuable insights regarding microbiome responses to FMT oral capsules in cats with chronic digestive issues.

Second, we did not carry out metagenomic sequencing, long-read sequencing or measure strain-level patterns when evaluating microbiome responses or determining bacterial engraftment rates. Our analyses were instead conducted at the level of bacterial ASVs, which is the most refined level after “bacterial species”. Future studies should carry out other types of sequencing to gain more insight into which exact bacterial species are shared between donors and recipients or change in abundance as a result of FMT.

Diet is a large determinant of the fecal microbiome in companion animals. Due to our small cohort size, our diet analyses were extremely granular but future studies should examine the specific brand of food and diet compositions (% fiber, % protein, % fat) on FMT fecal microbiome responses. Another study can take it further and investigate whether dietary interventions (e.g., a change to a veterinary diet) modulate a cat’s experience with FMT. Also, given that many of the cats that are targets for FMT may have different types of chronic enteropathies, additional studies need to be conducted to evaluate responses to FMT in cats with IBD, non-IBD enteropathy, IBD-non-lymphoma enteropathy, and IBD-lymphoma enteropathy.

Lastly, our fecal microbiome surveys reflect the fecal microbiome of FMT recipients at two weeks post-FMT. We do not know how the microbiome changed after those two weeks, nor can we comment on the long-term effects of FMT oral capsules or the possible need for further applications of the FMT capsules. Our analyses are focused on the fecal microbiomes of cats during a particular window and we encourage future studies to examine fecal microbiomes at distinct time windows post-FMT. More work needs to be conducted to determine the optimal dosage and duration of FMT which will depend on the individual cat.

We acknowledge that all authors that conducted this study have a conflict of interest in that they are employees or advisors to AnimalBiome, the company that produced the FMT capsules. Nonetheless, we have made every effort to be as transparent as possible by providing a highly detailed methods section, being explicit about the statistical analyses we conduct, and sharing our code and data. We do not shy away from pointing out our study’s limitations as well.

## 6. Conclusions

Our study shows fecal microbiome responses to FMT were potentially modulated by an individual’s clinical signs, and dry kibble consumption, and to a lesser extent by prior antibiotic use, which are all factors to consider when studying the fecal microbiomes of cats. Partial stool donor bacterial engraftment was observed in FMT recipients, illustrating that microbes are being shared between a stool donor and its recipient, and that complete microbiome engraftment may not be necessary for FMT to have an impact. The fecal microbiomes of Responders grew more similar to the microbiomes of age-matched and diet-matched healthy individuals, which is promising but future studies are required to disentangle the nuances regarding FMT and its impact on cat health. Our work is starting these conversations and adds to the small but growing body of work examining the effects of FMT in felines.

## Figures and Tables

**Figure 1 vetsci-10-00561-f001:**
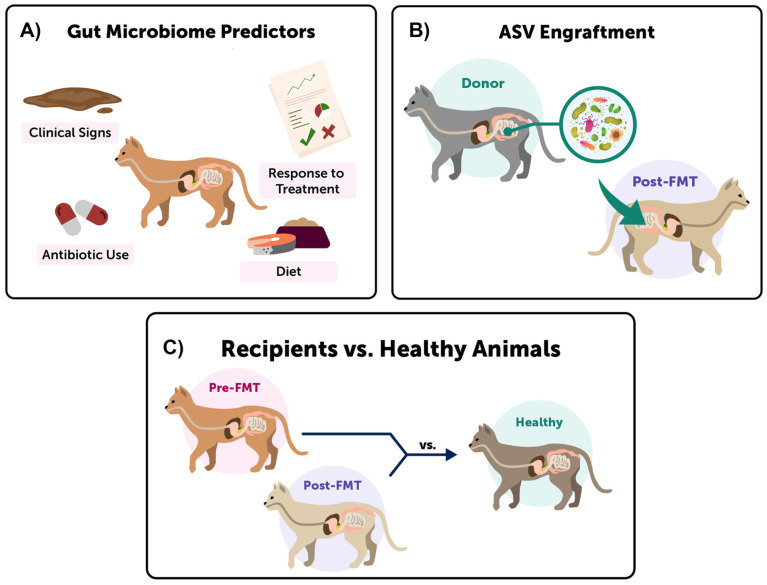
Three main fecal microbiome comparisons were conducted in this study. (**A**) We investigated shifts in the fecal microbiomes of FMT recipients by testing whether four host predictors were significantly associated with microbiome composition, alpha-diversity, and beta-diversity. (**B**) We identified which ASVs ‘engrafted’ in FMT recipients by comparing the fecal microbiomes of FMT recipients to those of their FMT stool donors, and determining which ASVs were shared. (**C**) Lastly, we examined whether the fecal microbiomes of FMT recipients became more similar to the microbiomes of healthy cats after FMT.

**Figure 2 vetsci-10-00561-f002:**
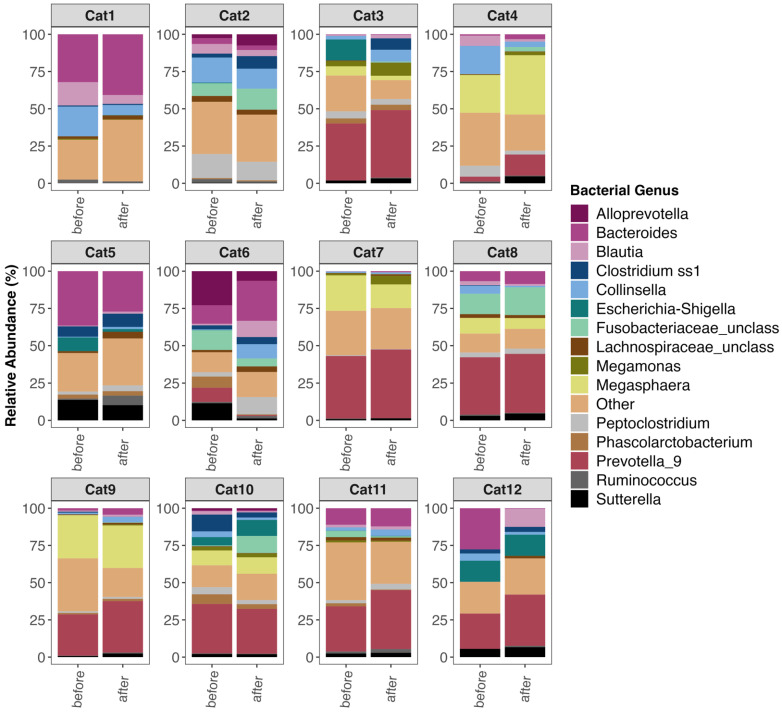
Fecal microbiome composition before and after FMT for twelve of the forty-six cats. Plots showing the relative frequencies of 16S rRNA gene sequences assigned to bacterial genera with mean relative abundances > 1.65%, while all other genera are clumped into an “Other” category. Twelve of the forty-six cats were selected at random for plotting. For plots of the microbiome compositions of all 46 cats, see Appendix A.

**Figure 3 vetsci-10-00561-f003:**
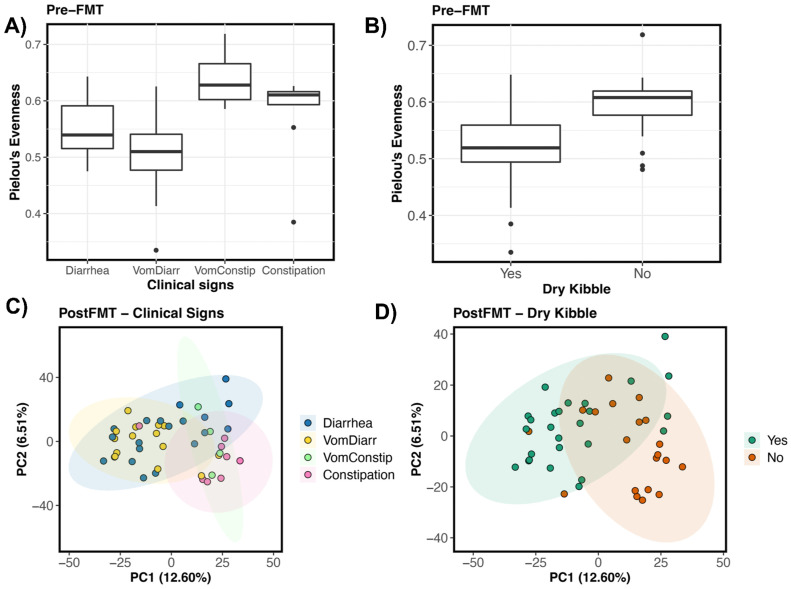
Host predictors of fecal microbiome alpha- and beta-diversity in FMT recipients. Boxplots of microbiome evenness (Pielou’s Evenness) by (**A**) clinical signs and (**B**) dry kibble consumption for pre-FMT samples. PCoA ordinations based on Aitchison distances showing the clustering of post-FMT samples by (**C**) Clinical signs, and (**D**) Dry kibble consumption.

**Figure 4 vetsci-10-00561-f004:**
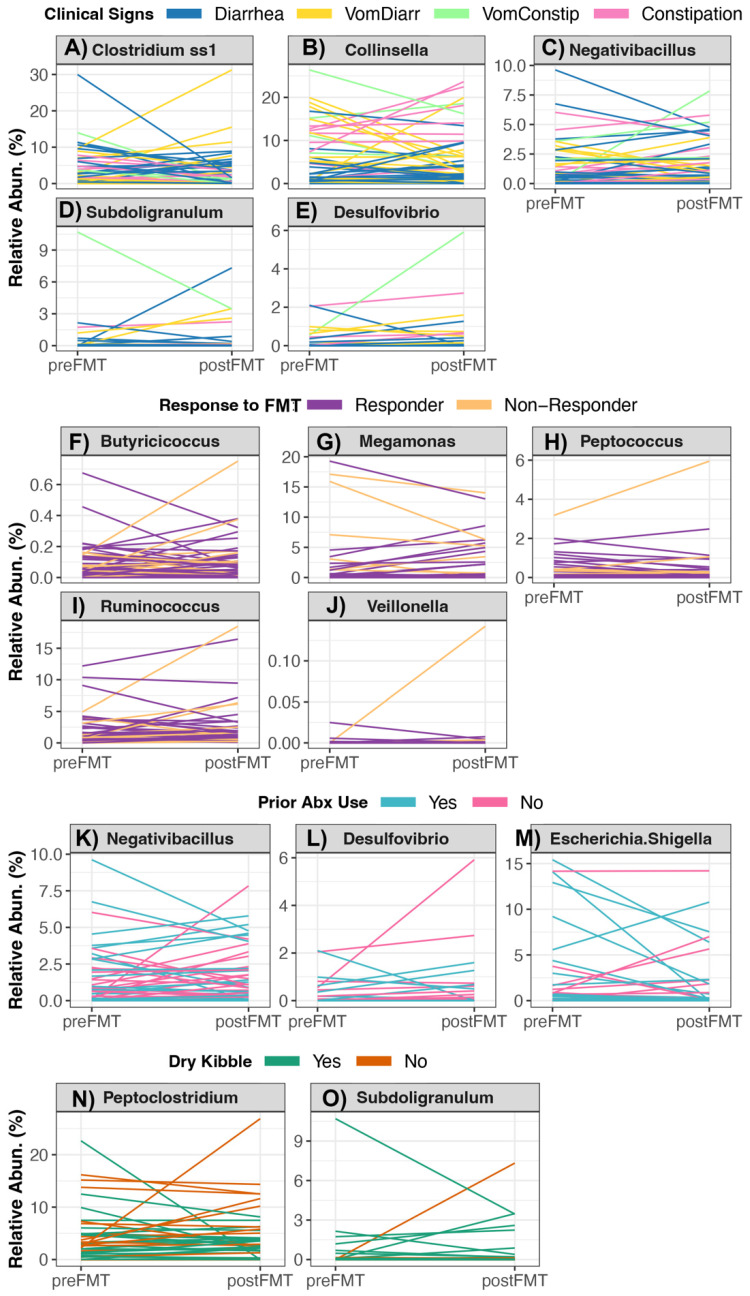
Changes in the relative abundances of bacterial genera in cats receiving oral FMTs. Linear mixed models indicated that the changes in the relative abundance of 10 bacterial genera were significantly associated with (**A**–**E**) host clinical signs, (**F**–**J**) response to FMT, (**K**–**M**) recent antibiotic use, or (**N**,**O**) dry kibble consumption. For the statistical output of post-hoc testing using Tukey linear contrasts, see Appendix A.

**Figure 5 vetsci-10-00561-f005:**
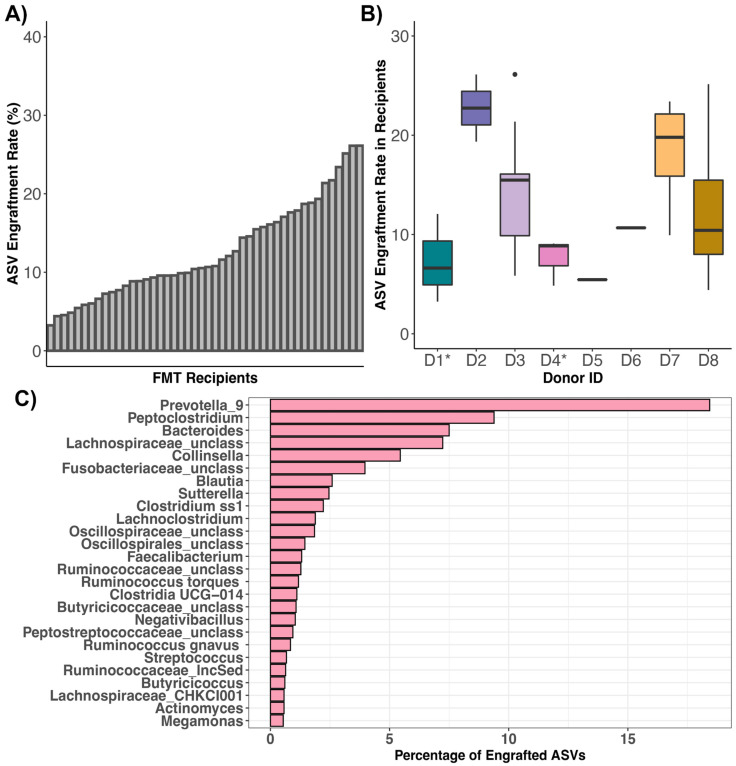
ASVs shared between FMT recipients and their stool donors. Plots of ASV engraftment or sharing rates across (**A**) FMT recipients or (**B**) stool donors (asterisks indicate where the fecal material of two cohabiting cats constituted one donor). ASV sharing rates were calculated by dividing the number of ASVs shared between postFMT samples of FMT recipients and their stool donors (excluding taxa shared between preFMT samples and donors) by the total number of ASVs in the donor sample (excluding any taxa shared with the preFMT samples of FMT recipients). (**C**) Taxonomic breakdown of ASVs shared between FMT recipients and their stool donors.

**Figure 6 vetsci-10-00561-f006:**
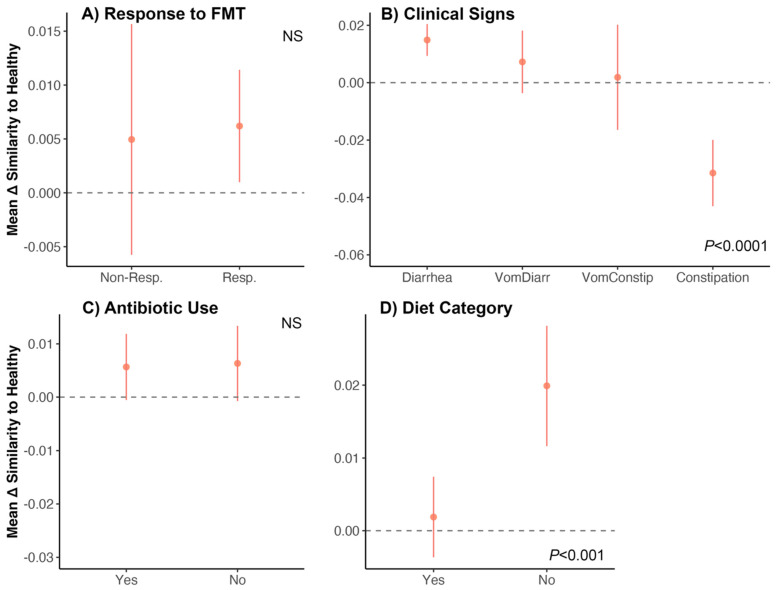
Do the fecal microbiomes of cats receiving FMT become similar to the microbiomes of healthy cats? We examined shifts in the fecal microbiome of FMT recipients by quantifying how similar their fecal microbiomes were (before and after FMT) to those from an age-matched and diet-matched healthy reference set. We computed the difference between the two similarity scores (postFMT similarity–preFMT similarity) to generate the change in similarity (Δ) for each FMT recipient–healthy animal dyad. We then correlated these Δ values with host characteristics using generalized linear models. (**A**–**D**) Average Δ in similarity scores for each group (±St.Error). A value of 0 indicates no shift in similarity. Bray–Curtis distances were used. See Appendix A for model statistics.

**Table 1 vetsci-10-00561-t001:** Summary characteristics for the forty-six cats that received oral capsule FMTs.

Characteristic	Subcategory	FMT Recipients (N = 46)
Age, in years	mean ± SD	10.23 ± 4.03
Body condition (1–10)	mean ± SD	4.72 ± 1.67
Sex	Female	20 (44%)
	Male	26 (56%)
Breed	Domestic Shorthair	34 (74%)
	Other breed	12 (26%)
Diet (not mutually exclusive)	Include Dry Kibble in their diet	26 (56%)
	Include Raw food in their diet	22 (48%)
	Include Canned Food in their diet	29 (63%)
Spayed or Neutered	Yes	46 (100%)
Antibiotics	Yes	24 (52%)
	No	22 (48%)
Initial clinical symptoms	Diarrhea only	19 (41%)
	Vomiting with Diarrhea	15 (33%)
	Vomiting with Constipation	4 (9%)
	Constipation only	8 (17%)

Cats exhibiting symptoms of a chronic digestive condition (e.g., diarrhea, vomiting, and/or constipation episodes lasting >2 weeks) were recruited for this study. Owners completed a health and demographic survey on their cats and collected fecal samples before and two-weeks after the end of a course of 50 FMT capsules.

## Data Availability

Raw amplicon sequences are available upon request by emailing the corresponding author. The ASV relative abundance table, ASV taxonomic classifications, and corresponding sample metadata are available as Appendix A. The R code for conducting all statistical analyses and generating all figures presented in this article is stored in a public GitHub repository https://github.com/AnimalBiome/MicrobiomeResponsestoFMT_inCats (accessed on 4 September 2023).

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
