# Peer review of "Microbiome Responses to Fecal Microbiota Transplantation in Cats with Chronic Digestive Issues"

_vetsci, 2023, doi:10.3390/vetsci10090561_

Round 1

Reviewer 1 Report

The strength of this manuscript is exactly what you would expect from a commercial entity – the breadth and depth of diagnostic investigation into the specifics of the FMT and microbiome are impressive.  The study falls apart as an attempt to run any sort of “clinical trial” or support the use of fecal capsules as a viable therapeutic option.  The difference between the two attempts is even glaring clear in the difference in the writing – the clinical argument is sloppy and poorly presented while the actual scientific methods and efforts are quite well written. As I state in my conclusion, the way to save this manuscript is to cut out the clinical attempt (recruit and fund an unbiased academic institution to design and run a clinical trial with no publication constraints) and stick to the science.

Some specifics.

Simple Summary still contains (76%) while Abstract states 76%

Table 1. Summary…for forty-eight cats… but then in the table itself (N=46)

I would remove any language anywhere in the manuscript (including Tables) of “suspected IBD”.  If you want to analyze a separate and distinct subset of cats it should be those cats with a histopathologic diagnosis of IBD in some part of their intestinal tract (also emphasizing that even this group may include GI Lymphoma cats because of the limitations discussed in previous review).  Suspected and strongly suspected IBD cats are NOT IBD CATS without evidence of a failed 2-week hypoallergenic/hydrolyzed diet trial and histopathology (without a diet trial as a diagnostic test for food allergy, the histopathology from a food allergy cat will look very much like an IBD cat); these are simply Chronic enteropathy cats.

Line 146   “…typical fecal consistency of their cat…” needs to be changed, the consistency refers to the feces, not the cat

Line 148  “hard” refers to the feces, but “constipated” refers to a clinical condition of the cat, it may be very inaccurate to assume hard feces results in a constipated cat.

Line 160  “…consistency cores” (scores)

Line 360  as stated above, without a failed diet trial and histopathology, cats can be “given a diagnosis of IBD” but NOT diagnosed with IBD.  Hell, I can “give” your cat any diagnosis you want me to, as long as you don’t care if it’s not the correct diagnosis. What is the obsession with IBD?  Why not keep this a study of cats with Chronic Enteropathy (as defined in the manuscript, although this will be the first time CE is defined as “clinical signs greater than 1 week” – this was a sloppy mistake, it is abundantly clear to anyone paying attention to the veterinary literature that the absolute MINIMUM to qualify as Chronic Enteropathy is 2-weeks of clinical signs, often more.

Line 368  this is a confusing distinction, these cats have chronic GI problems but “were stable” at the beginning of the study?  So the signs had resolved? The signs hadn’t appeared in the previous 1 week, previous 24 hours? The cats were not in-patients for some other reason?  What (and why) do you mean by bringing this up?  That also makes the final sentence very confusing, “Whether this changed…”  What is “this”, their stable state? The chronic state? Are these two states different?  

Line 383  50% of cats had recently taken antibiotics???  That’s a problem.

Line 394: no place for “a cursory glance” in a scientific manuscript

Line 408: Stop it.  This is not a dinner party conversation, it’s a Results section; “To better parse out this variation…”.  The correlation between XXX and YYY was determined using ZZZ.

Line 415: “…less even fecal…”

Tons of misleading statements are being made in the RESULTS section of the manuscript. All of them need to either be identified as statistically significant with the appropriate p values or clearly identified as not reaching statistical significance.   

Line 553  missing x:

Figure 6.  A) Response to Tmt…what is Tmt?  If I am correct that this is supposed to be “Response to FMT” then it is a bit embarrassing

Results are chocked full of these sorts of statements

            Response to FMT was not significantly associated with fecal microbiome variation.

             Generally…tended to increase…

            Tended to decrease slightly…

            Tended to increase…

            Tended to showcase…

            Exhibited decrease…(again, no statistical justification for this statement)

Discussion

The discussion, like the rest of the paper, is of “two minds”.  The attempt to present the clinical consequences of FMT is a mess, starting with the Perceived effectiveness section, with the most useful message being that one size does not fit all.

Was there a significant decrease in the fecal score for the Responder cats, irrespective of that comparison with Non-Responders?  i.e. making the comparison between those two groups introduces a bias that is strongly in favor of the two fecal scores being significantly different…almost by definition that’s going to be the case, but means very little – I want my cat’s feces to improve, I don’t care what it’s doing compared to my neighbor’s cat.  And don’t both groups include cats with constipation?  That’s a mess; you have groups where you want some cats’ feces to get more firm while you want other cats within the same group to have their feces get softer.

After that, the discussion of the microbiome and FMT, starting with Host predictors, is one of the best discussions I’ve read in a long time; clear, informative, specific and succinct.

To save this paper I would stop trying to sell the capsule or even discuss any clinical impact in these wacko groups, and present what you do best, which is the thorough, difficult, complex, but essential characterization of FMTs, recipients, donors, and relationships between the two.  That would also completely remove any conflict of interest (capsule FMT coming from these donors does this, this, and this to the microbiome of cats with chronic enteropathy). 

I’d give you one and only one sales pitch;

76% (or if you prefer, approximately 76%) of owners reported an improvement in chronic enteropathy signs following a 50-treatment schedule of capsule FMT, while 24% reported no change or worsening of clinical signs. (the fecal argument is biased and bogus, leave it out). This qualitative result is clearly subject to bias but supports the need for well-designed prospective studies of FMT in well-defined groups of CE cats using validated and quantifiable measures of clinical response.

[Otherwise, fund an academic institution to design and run a clinical trial where you provide the capsules but have no other control of the study, the results, or the manuscript.]

The rest of your manuscript would be an important addition to the field of FMT.

Reviewer 2 Report

Dear authors

Thank you for considering my comments of the first review round and for revising the manuscript accordingly. From my point of view, the data presentation and interpretation have been markedly improved, why I would like to recommend the acceptance of the manuscript for publication.

Only one very small comment:

Lines 614-616: “The type of food a cat is eating (…) depending on its protein, fat, carbohydrate, fiber, and macronutrient composition [61–64].”

As protein, carbohydrates and fat are macronutrients, the wording “and macronutrient composition” is a bit misleading. Please rephrase this sentence before or during proofreading.

Reviewer 3 Report

The authors present a manuscript addressing a topic that I consider highly significant within the field of veterinary medicine, particularly for its future, namely the microbiome. As the authors delineate at the outset of the abstract, fecal microbiota transplants (FMTs) constitute an approach that has been witnessing a growing application in veterinary medicine, drawing the attention of numerous animal owners. Nonetheless, this approach still lacks comprehensive scientific data.

I hold the manuscript in high scientific esteem, particularly for demonstrating the efficacy of FMT. However, I do have a few suggestions:

Between lines 86 and 99 of the introduction, there are pieces of information that, in my opinion, could be better situated within the "Materials and Methods" section. This is especially true for those references to the table and figures. These details might be succinctly summarized within the introduction and more elaborately elaborated upon in the "Materials and Methods" section. It appears to me that they are somewhat misplaced within the introduction.

Furthermore, I found it somewhat lacking that the authors did not expound on the advantages and disadvantages of FMT as compared to probiotics and prebiotics.

Furthermore, I would be more than pleased to endorse the publication of this manuscript as an article, provided my queries are addressed. I would like to extend my congratulations to the authors for this engaging work.

Round 2

Reviewer 1 Report

I think the authors have appropriately "repurposed" the presentation of their study and it has resulted in a much improved and impactful paper.